# Two Sides of The Same Coin:
# Bridging Deep Equilibrium Models and Neural ODEs via Homotopy Continuation

**Shutong Ding**[*]
ShanghaiTech University
dingsht@shanghaitech.edu.cn

**Tianyu Cui**[*]
ShanghaiTech University
cuity2022@shanghaitech.edu.cn

**Jingya Wang**
ShanghaiTech University
wangjingya@shanghaitech.edu.cn

**Ye Shi** [†]
ShanghaiTech University
shiye@shanghaitech.edu.cn

## Abstract

Deep Equilibrium Models (DEQs) and Neural Ordinary Differential Equations (Neural ODEs) are two branches of implicit models that have achieved remarkable success owing to their superior performance and low memory consumption. While both are implicit models, DEQs and Neural ODEs are derived from different mathematical formulations. Inspired by homotopy continuation, we establish a connection between these two models and illustrate that they are actually two sides of the same coin. Homotopy continuation is a classical method of solving nonlinear equations based on a corresponding ODE. Given this connection, we proposed a new implicit model called HomoODE that inherits the property of high accuracy from DEQs and the property of stability from Neural ODEs. Unlike DEQs, which *explicitly* solve an equilibrium-point-finding problem via Newton's methods in the forward pass, HomoODE solves the equilibrium-point-finding problem *implicitly* using a modified Neural ODE via homotopy continuation. Further, we developed an acceleration method for HomoODE with a shared learnable initial point. It is worth noting that our model also provides a better understanding of why Augmented Neural ODEs work as long as the augmented part is regarded as the equilibrium point to find. Comprehensive experiments with several image classification tasks demonstrate that HomoODE surpasses existing implicit models in terms of both accuracy and memory consumption.

## 1 Introduction

Recent studies of implicit models have certified that such models can meet or even surpass the performance of traditional deep neural networks. Instead of specifying the explicit computation process of the output, implicit models define the joint conditions that the layer's input and output satisfy. Instances of such models include the Neural Ordinary Differential Equations (Neural ODEs) [10], which treat ODEs as a learnable component and can be viewed as continuous Residual Networks [25] (ResNets), the Deep Equilibrium Models (DEQs) [6], which compute the equilibrium point of a nonlinear transformation corresponding to an infinite-depth weight-tied network; and optimization layers [4, 45], which leverage the optimization techniques as layers of neural networks.

Although DEQs and Neural ODEs, as popular implicit models in recent years, have garnered much attention in terms of theoretical analysis and application, an insightful connection between these two

---

[*]Equal contribution. [†]Corresponding author.

37th Conference on Neural Information Processing Systems (NeurIPS 2023).

branches of the implicit model has not been established. DEQs involve an equilibrium-point-finding problem, which is a nonlinear equation system. Generally, this can be solved via the homotopy continuation method [39], a classical method that solves nonlinear equations along the zero path of the homotopy mapping and can be further formulated as an ODE. This motivated us to consider whether we could bridge DEQs and Neural ODEs via the theory of homotopy continuation.

In this paper, we show that Neural ODEs can also be viewed as a procedure for solving an equilibrium-point-finding problem. However, while both of the two models can be considered as solving equilibrium-point-finding problems, they differ in how the input information is used. On the one hand, DEQs regard the input information as the *condition* that determines the equilibrium-point-finding problem to solve via injecting it into the equilibrium function in the forward pass. On the other hand, Neural ODEs generate the initial points with different input information and expect them to converge to different equilibrium points. Therefore, we claim that DEQs and Neural ODEs are actually two sides of the same coin.

Inspired by the theoretical connection between the two models, we developed a new implicit model called HomoODE, which inherits the advantages of both. Specifically, HomoODE injects the input information into the underlying dynamic of an equilibrium-point-finding problem in the same way as a DEQ does, but then obtains the output from an ODE solver just as a Neural ODE does. The connection between DEQ and Neural ODE means that HomoODE avoids DEQ's problem of unstable convergence in DEQs and the weak representation capability of Neural ODEs. Further, a common issue for implicit models is the trade-off between computational speed and precision. Therefore, a natural intuition to accelerate such models is to find a good initial point that is close to the solution. We observed that, as the distance between the initial point and the solution drops to some range, the numbers of function evaluations (NFE) almost stop decreasing when applying homotopy continuation. Hence, we introduced an extra loss instead of zero vector initialization in DEQ to train a shared initial point. In this way, the distance from the initial point to each solution is within appropriate ranges. A series of experiments with several image classification tasks verify that HomoODE is able to converge stably and that it outperforms existing implicit models with better memory efficiency on several image classification tasks, and our acceleration method significantly reduces the NFE of HomoODE. In summary, our contributions are as follows:

(1) **A connection between DEQs and Neural ODEs.** We establish a connection between DEQs and Neural ODEs via homotopy continuation, which illustrates that DEQs and Neural ODEs are actually the two sides of the same coin. We believe this new perspective provides novel insights into the mechanisms behind implicit models.

(2) **A New Implicit Model: HomoODE.** We propose a new implicit model called HomoODE, that inherits the advantages of both DEQs and Neural ODEs. HomoODE *implicitly* solves equilibrium-point-finding problems using homotopy continuation, unlike DEQs which *explicitly* solve these problems via Newton's methods. Additionally, we have accelerated HomoODE with a learnable initial point that is shared among all samples.

(3) **Understanding Augmented Neural ODE.** We demonstrate that Augmented Neural ODE can be treated as a special case of HomoODE based on homotopy continuation. Hence Augmented Neural ODE enjoys better representation ability and outperforms Neural ODE.

(4) **Better Performance.** We conduct experiments on image classification datasets and confirm our HomoODE outperforms DEQ, Neural ODE, and variants of them both in accuracy and memory usage. Furthermore, we also perform the sensitivity analysis on the hyper-parameters to research the characters of our model.

## 2   Related Works

**Deep Equilibrium Models**. DEQs [6] have shown competitive performance on a range of tasks, such as language modeling[6], graph-related tasks [20], image classification or segmentation [7], image generation [41], inverse problems in imaging[16], image denoising [17] and optical flow estimation [5]. DEQs find an equilibrium point of a nonlinear dynamical system corresponding to an effectively infinite-depth weight-tied network. However, training such models requires careful consideration of both initializations and the model structure [6, 9, 3], and often consumes long training times. Many studies have been devoted to solving these problems. For example, the Monotone Operator

Equilibrium Network (monDEQ) [46] ensures stable convergence to a unique equilibrium point by involving monotone operator theory. Bai et al. [9] propose an explicit regularization scheme for DEQs that stabilizes the learning of DEQs by regularizing the Jacobian matrix of the fixed-point update equations. Kawaguchi et al.[30] prove that DEQs converge to global optimum at a linear rate for a general class of loss functions by analyzing its gradient dynamics. Agarwala et al.[3] show that DEQs are sensitive to the higher-order statistics of their initial matrix family and consequently propose a practical prescription for initialization. To reduce the computational expense of DEQ, Pal et al. [40] developed continuous DEQs utilizing an "infinity time" neural ODE but did not continue to explore the inherent connection between general DEQs and Neural ODEs. From the perspective of optimization, Optimization Induced Equilibrium Networks (OptEq)[47] theoretically connect their equilibrium point to the solution of a convex optimization problem with explicit objectives. Instead of regularizing the structures or involving parameterizations of the implicit layer design, Bai et al. [8] propose a model-specific equilibrium solver, which both guesses an initial value of the optimization and performs iterative updates. However, unlike DEQs, which *explicitly* solve equilibrium-point-finding problems via Newton's methods, our HomoODE solves these problems based on homotopy continuation *implicitly*. Accordingly, HomoODE does not suffer from the issue of unique equilibrium like DEQ and thus can avoid the stability issue. Moreover, we accelerate HomoODE using a good initial point learned with a corresponding loss function. Unlike Bai et al.'s approach [8], HomoODE learns an initial point shared among all samples without involving a network-based initializer.

**Neural Ordinary Differential Equations**. Neural ODEs have been applied to time series modeling [44, 32], continuous normalizing flows [10, 18], and modeling or controlling physical environments [50, 43, 48, 19, 14]. Neural ODEs treat ODEs as a learnable component and produce their outputs by solving the Initial Value Problem [10, 12, 29]. However, Neural ODEs are often characterized by long training times and sub-optimal results when the length of the training data increases [13, 15]. Prior works have tried to tackle these problems by placing constraints on the Jacobian[13] or high derivatives of the differential equation [31]. Conversely, Augmented Neural ODEs [12] learn the flow from the input to the features in an augmented space with better stability and generalization. Ghosh et al. [15] treat the integration time points as stochastic variables without placing any constraints. With diffeomorphism, the complexity of modeling the Neural ODEs can be offloaded onto the invertible neural networks [49], and training Neural ODEs with the adaptive checkpoint adjoint method [51] can be accurate, fast, and robust to initialization. The symplectic adjoint method [36] finds the exact gradient via a symplectic integrator with appropriate checkpoints and memory consumption that is competitive to the adjoint method. The advantage of HomoODE is that it inherits the property of high accuracy from DEQs and the property of stability from Neural ODEs. In addition, HomoODE provides an explanation of why Augmented Neural ODEs achieve better performance than Neural ODEs. Notably, Augmented Neural ODEs can be viewed as a special case of HomoODE.

**Homotopy Continuation.** Homotopy continuation [39] is a numerical technique that traces the solution path of a given problem as the parameter changes from an initial value to a final value. Homotopy methods have been successfully applied to solving pattern formation problems arising from computational mathematics and biology including computing multiple solutions of differential equations [21, 22], state problems of hyperbolic conservation laws [21], computing bifurcation points of nonlinear systems [24] and solving reaction–diffusion equations [23]. Recent advances in deep learning have also seen the homotopy continuation method fused into learning processes. For instance, Ko et al.[33] adapt homotopy optimization in Neural ODEs to gain better performance with less more training epochs. HomPINNs [27] traces observations in an approximate manner to identify multiple solutions, then solves the inverse problem via the homotopy continuation method. To reach a good solution to the original geometrical problem, Hruby et al. [26] learn a single starting point for a real homotopy continuation path. In this work, we establish a connection between DEQs and Neural ODEs from the perspective of homotopy continuation and develop a new implicit model called HomoODE based on this theoretical relationship.

## 3   Background on Homotopy Continuation

Homotopy continuation has been broadly applied to solve nonlinear equations. The first step to solving a specific problem $r(z) = 0$ is to construct a homotopy mapping.

**Definition 1** *(Homotopy mapping [39]) The function $H(z, \lambda) = \lambda r(z) + (1 - \lambda)g(z)$ is said to be a homotopy mapping from $g(z)$ to $r(z)$, if $\lambda$ is a scalar parameter from $0$ to $1$, and $g(z)$ is a smooth function. The equation $H(z, \lambda) = 0$ is the zero path of this homotopy mapping.*

Homotopy mapping provides a continuous transformation by gradually deforming $g(z)$ into $r(z)$ while $\lambda$ increases from $0$ to $1$ in small increments. Hence, the solution to $r(z)$ can be found by following the zero path of the homotopy mapping $H(z, \lambda) = 0$. Usually, one can choose $g(z)$ as an artificial function with an easy solution. Here, we specifically consider Fixed Point Homotopy which chooses $g(z) = z - z_0$:

$$H(z, \lambda) = \lambda r(z) + (1 - \lambda)(z - z_0), \tag{1}$$

where $z_0 \in \mathbb{R}^n$ is a fixed vector, and is the initial point of the homotopy continuation method. In one practical trick, we can follow the zero path by allowing both $z$ and $\lambda$ to be functions of an independent variable $s$, which represents arc length along the path. In other words, $(z(s), \lambda(s))$ is the point arrived at by traveling a distance $s$ along the zero path from the initial point $(z(0), \lambda(0)) = (z_0, 0)$. In the zero path, we have $H(z(s), \lambda(s)) = 0$, for all $s \geq 0$.

Take the derivative for this equation with respect to $s$ lead to:

$$\frac{\partial H(z, \lambda)}{\partial z}\frac{dz}{ds} + \frac{\partial H(z, \lambda)}{\partial \lambda}\frac{d\lambda}{ds} = 0. \tag{2}$$

The vector $(\frac{dz}{ds}, \frac{d\lambda}{ds}) \in \mathbb{R}^{n+1}$ is the tangent vector to the zero path, and it lies in the null space of matrix $\left[\frac{\partial H(z,\lambda)}{\partial z}, \frac{\partial H(z,\lambda)}{\partial \lambda}\right] \in \mathbb{R}^{n \times (n+1)}$. To complete the definition of $(\frac{dz}{ds}, \frac{d\lambda}{ds})$, a normalization condition is imposed to fix the length of the tangent vector, i.e.

$$\left\|\frac{dz}{ds}\right\|^2 + \left|\frac{d\lambda}{ds}\right|^2 = 1. \tag{3}$$

Given the tangent vector and the initial point, we can trace the zero path and obtain the solution of $F(z) = 0$ by solving the ODE (2).

## 4 Bridging DEQs & Neural ODEs via Homotopy Continuation

Here we briefly review DEQs and Neural ODEs, then bridge these two models via homotopy continuation. DEQs aim to solve the equilibrium point of the function $f(z; x, \theta)$, which is parameterized by $\theta$ and the input injection $x$. The underlying equilibrium-point-finding problem of DEQs is defined as follows:

$$z^\star = f(z^\star; x, \theta). \tag{4}$$

Usually, we choose $f(z; x, \theta)$ as a shallow stacked neural layer or block. Hence, the process of solving the equilibrium point can be viewed as modeling the "infinite-depth" representation of a shallow stacked block. One can use any black-box root-finding solver or fixed point iteration to obtain the equilibrium point $z^\star$.

Unlike the underlying equilibrium-point-finding problem of DEQs, Neural ODEs view its underlying problem as an ODE, whose derivative is parameterized by the network. Specifically, Neural ODEs map a data point $x$ into a set of features by solving the Initial Value Problem [29] to some time $T$. The underlying ODE of Neural ODEs is defined as follows:

$$\frac{dz(t)}{dt} = F(z(t), t; \theta), \quad z(t_0) = x, \tag{5}$$

where $z(t)$ represents the hidden state at time $t$, $F(z(t), t; \theta)$ is neural networks parameterized by $\theta$.

**The same coin.** DEQs apply Newton's methods to solve the underlying equilibrium-point-finding problem $z = f(z; x, \theta)$. By defining $r(z) = z - f(z; x, \theta)$, one can alternatively solve this equilibrium-point-finding problem based on homotopy continuation. Now we will show that the underlying dynamics in Neural ODEs can also be treated as an equilibrium-point-finding problem.

Firstly, we apply the Fixed Point Homotopy to solve the equilibrium-point-finding problem $z = f(z; \theta)$ and obtain the homotopy mapping $H(z, \lambda) = \lambda(z - f(z; \theta)) + (1 - \lambda)z$. Taking the partial derivative of $H(z, \lambda)$ with respect to $z$ and $\lambda$, respectively, we obtain

$$\frac{\partial H(z, \lambda)}{\partial z} = I - \lambda \nabla_z f(z; \theta), \quad \frac{\partial H(z, \lambda)}{\partial \lambda} = -f(z; \theta). \tag{6}$$

By substituting the partial derivative in (6) into (2), we can obtain:

$$\frac{dz}{ds} = (I - \lambda \nabla_z f(z;\theta))^{-1} f(z;\theta) \frac{d\lambda}{ds}. \tag{7}$$

Based on the normalization condition (3), we can reformulate (7) as the following differential equation:

$$\frac{dz}{ds} = (I - \lambda(z)\nabla_z f(z;\theta))^{-1} f(z;\theta)\sqrt{1 - \left\|\frac{dz}{ds}\right\|^2}. \tag{8}$$

As Neural ODEs do, we can use neural networks to approximate the underlying dynamics of such an ODE (8). However, the norm of neural network output is likely to exceed the unit length, i.e. violating the normalization condition (3). To address this issue, we introduce $v := \frac{ds}{dt}$ as the velocity of the point $(z, \lambda)$ traveling along the zero path, and modify the normalization condition by introducing $v$ into (3):

$$\left\|\frac{dz}{dt}\right\|^2 + \left|\frac{d\lambda}{dt}\right|^2 = v^2. \tag{9}$$

Note that the convergence of the homotopy continuation is not affected by the value of $v$. The underlying dynamics of (7) becomes

$$\frac{dz}{dt} = (I - \lambda(z)\nabla_z f(z;\theta))^{-1} f(z;\theta)\sqrt{v^2 - \left\|\frac{dz}{dt}\right\|^2}. \tag{10}$$

Following Neural ODEs, the differential equation (10) can be approximated by neural networks, i.e. $\frac{dz}{dt} = F(z(t), t; \theta)$. However, we still need to ensure the existence of a corresponding equilibrium-point-finding problem for Neural ODE. Ingeniously, the modified normalization can also ensure the existence of the equilibrium-point-finding problem. When we obtain the Neural ODEs (5) by training the neural networks $F(z(t), t; \theta)$, we can compute the changing process of $\lambda(t)$ and the velocity $v$ by solving the following equations:

$$\frac{d\lambda}{dt} = \sqrt{v^2 - \|F(z(t), t; \theta)\|^2}, \quad \lambda(0) = 0, \quad \lambda(1) = 1. \tag{11}$$

Note that the modified normalization condition (9) provides the dynamic with another degree of freedom, which guarantees the existence of $\lambda(t)$. Otherwise, there might be no solution for $\lambda(t)$ as there are two initial conditions (i.e., $\lambda(0) = 0, \lambda(1) = 1$) for the system. Hence, the equilibrium-point-finding problem $z = f(z; \theta)$ is implicitly determined by the following partial differential equation:

$$F(z(t), t; \theta) = (I - \lambda(t)\nabla_z f(z;\theta))^{-1} f(z;\theta)\sqrt{v^2 - \|F(z(t), t; \theta)\|^2}. \tag{12}$$

Therefore, Neural ODEs can be regarded as the procedure of solving an equilibrium-point-finding problem with homotopy continuation, and the hidden state at $t = 0$, $z(t_0)$ is the initial point of homotopy continuation.

**Two sides.** We have shown that both DEQs and Neural ODEs can be considered as solving equilibrium-point-finding problems through homotopy continuation. Now we discuss the difference in underlying equilibrium-point-finding problem between DEQs and Neural ODEs.

On the one hand, DEQs solve the problem from the same initial point $z_0$. The equilibrium-point-finding problem of DEQs is parameterized by the input injection $x$. The underlying equilibrium-point-finding problem of DEQs is defined as follows:

$$z^\star = f(z^\star; x, \theta), z^{(0)} = z_0 \tag{13}$$

The input injection $x$ can be viewed as the *condition* to fuse the information of input to the underlying equilibrium-point-finding problem. The underlying problem of DEQs varies depending on different *conditions* $x$. Therefore, DEQs are able to map inputs to diverse representations, which is crucial for achieving superior performance.

On the other hand, unlike DEQs, Neural ODEs solve an equilibrium-point-finding problem with different initial points $z(t_0)$. The underlying equilibrium-point-finding problem of Neural ODEs is defined as follows:

$$z^\star = f(z^\star; \theta), \tag{14}$$

and Neural ODEs solve the problem through homotopy continuation:

$$\frac{dz(t)}{dt} = F(z(t), t; \theta), z(t_0) = x \tag{15}$$

where $f$ is determined by $F$ in equation (12). Concretely, Neural ODEs map raw data into a set of features $z(t_0) = x$ and regard them as the initial points of the ODE. The fixed underlying problem ensures the stability of Neural ODEs but loses diversified representation capabilities. Therefore, we claim that DEQs and Neural ODEs are actually two sides of the same coin from the perspective of homotopy continuation.

## 5   HomoODE: an efficient and effective implicit model

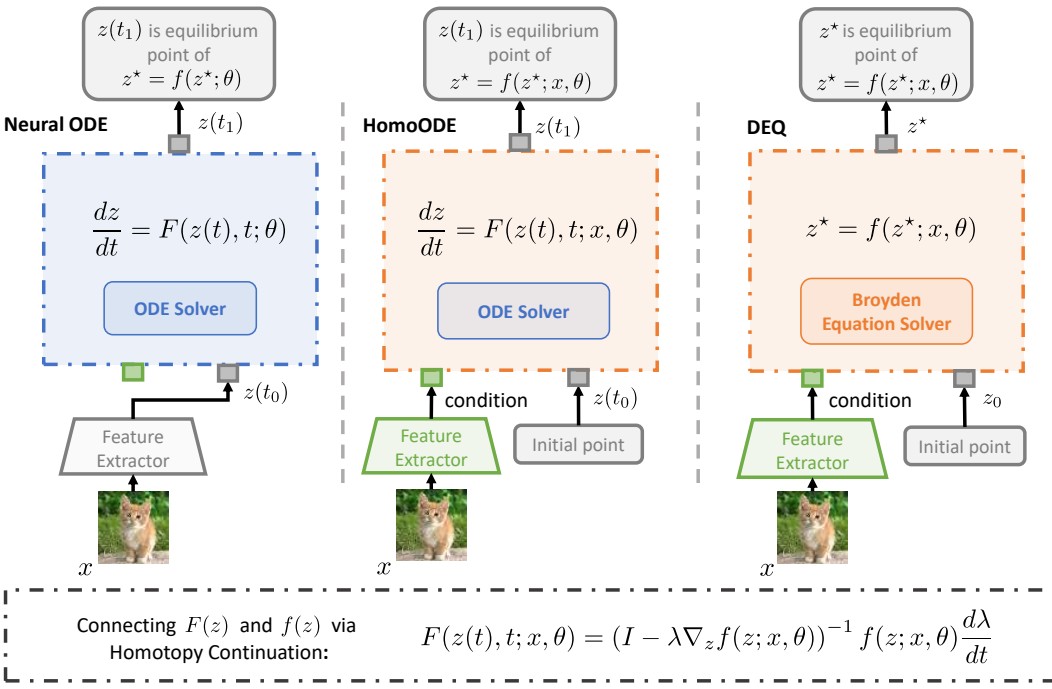

Figure 1: Comparison between different running mechanisms of implicit models.

As we show above, both DEQs and Neural ODEs can be considered as solving equilibrium-point-finding problems through homotopy continuation. Two well-known approaches for solving nonlinear equations are Newton's Method [2] and homotopy continuation [39]. DEQs solve an equilibrium-point-finding problem $r(z) = 0$ via Newton's Methods which are local in the sense that a good estimate of the solution is required for the convergence. Unlike Newton's Method, homotopy continuation is global in the sense that solutions of $g(z) = 0$ may not need to be anywhere close to the solution of $r(z) = 0$ [11]. Inspired by the connection between DEQs and Neural ODEs, a very natural thought is that we can apply homotopy continuation to solve the underlying nonlinear equation of DEQs. By replacing the equilibrium-point-finding problem $z = f(z; \theta)$ to $z = f(z; x, \theta)$, we can obtain the following differential equation:

$$\frac{dz}{dt} = (I - \lambda(z)\nabla_z f(z; x, \theta))^{-1} f(z; x, \theta) \sqrt{v^2 - \left\|\frac{dz}{dt}\right\|^2} \tag{16}$$

Hence, we proposed a new implicit model called HomoODE, which models an equilibrium problem $z = f(z; x, \theta)$ implicitly. Specifically, we employ neural networks to approximate the differential equation (16). In the design of the network structure, we introduce the *condition* $x$ into the underlying dynamic of HomoODE as DEQ does and obtain the output from the same initial point through the ODE solver as Neural ODE does. In this way, HomoODE not only has the ability of diversified

representation of DEQs but also has the property of stable convergence of Neural ODEs. In addition, the time information $t$ is not explicitly formulated by the dynamic of HomoODE (16). So unlike Neural ODEs, HomoODE does not require the input of time information $t$. Figure 1 illustrates the structure of HomoODE as well as Neural ODE and DEQ. Besides, it is worth noting that $v$ mentioned in (16) is just an auxiliary variable for the theoretical analysis. Therefore, we do not need to include it as a hyperparameter or compute its value in the forward pass because it has been implicitly contained in the neural network.

**Forward Pass.** In HomoODE, the raw data $x$ is first input to a feature extractor $g(x; \omega)$ and then injected into an ODE solver. Suppose $z(t)$ represents the intermediate state of HomoODE, calculating $z(t)$ involves an integration starting from the initial point $z(t_0) = 0$ to the solution $z(t_1)$. Notably, the output $z(t_1)$ is equivalent to the solution $z^\star$ of the implicit equilibrium-point-finding problem $z = f(z; x, \theta)$. Then we can use the ODE solvers to obtain the solution $z^\star$ of the origin equilibrium problem and this representation can be used for downstream tasks, such as classification, regression, etc.

$$z(t_1) = \text{ODESolve}(z(t_0), F(z(t), t; x, \theta), t_0, t_1) \tag{17}$$

**Backward Pass.** In the backward pass of HomoODE, we can apply the adjoint sensitivity method, or straightly differentiate through the operations of the forward pass. The *condition* traces back to another gradient flow. More details related to the construction of HomoODE dynamics and the computation of the gradients based on the adjoint method can be referred to in the supplementary materials.

## 6 Acceleration for HomoODE

In practice, we observe that HomoODE needs more function evaluations in the ODE-solving process, which results from bad initialization (e.g., zero vector initialization). To address this issue, we referred to [8], which makes an input-based initial point guess with an extra neural network to accelerate the equation-solving procedure in DEQ. However, this method poses an extra cost in both memory and computation.

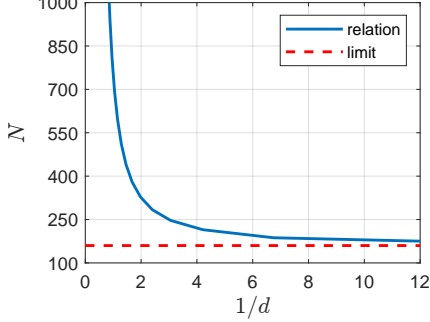

Figure 2: Relationship between the distance from $z_0$ to $z^\star$ and the iteration number of ODE solver. The x-axis is the inversion of the distance $1/d$, and the y-axis is the iteration number $N$.

To avoid these drawbacks, we further investigate the relationship between the number of iterations and the distance from the initialization $z_0$ to the solution $z^\star$ in homotopy continuation. Specifically, we perform the homotopy continuation method on a nontrivial nonlinear equation using ode45 in Matlab [28]. As is shown in Figure 2, it does not bring any reduction in the number of iterations when the initial point is close enough to the equilibrium solution. This means it is unnecessary to approximate the specific initial point very accurately for a specific sample $x$ in HomoODE, and that is different from that in [8].

Hence, we share the initial point for all the samples $x$ and just maintain a scalar value for the averaged value of the whole feature map in one channel. Briefly, we just need to store a tensor with the shape of $(1, 1, c)$ as the shared initial information $\tilde{z}_0$, and broadcast it into the initial point $z_0$ with the shape of $(h, w, c)$ when taking it into the ODE solver. Here $h, w, c$ means the height, width, and channel number of the feature map. The loss function of $\tilde{z}_0$ is defined as follows:

$$\mathcal{L}(\tilde{z}_0) := \mathbb{E}_{x \sim \mathcal{D}} \left[ (z^\star(x) - \tilde{z}_0)^2 \right], \tag{18}$$

where $z^\star(x)$ denotes the equilibrium solution of the sample $x$ and $\mathcal{D}$ denotes the distribution of $x$. In fact, this update on $\tilde{z}_0$ is equivalent to maintaining the dynamic geometrical center of the equilibrium points of all the samples.

## 7 Understanding Augmented Neural ODE by HomoODE

Augmented Neural ODEs, as a simple extension of Neural ODEs, are more expressive models and outperform Neural ODEs. Augmented Neural ODEs allow the ODE flow to lift points into the extra

dimensions to avoid trajectories crossing each other [12]. However, more theoretical analysis of how and why augmentation improves Neural ODEs is lacking. Our work provides another perspective on understanding the effectiveness of Augmented Neural ODEs. Augmented Neural ODEs formulate the augmented ODE problem as:

$$\frac{d}{dt}\begin{bmatrix}h(t)\\a(t)\end{bmatrix} = F(\begin{bmatrix}h(t)\\a(t)\end{bmatrix}, t; \theta), \qquad \begin{bmatrix}h(0)\\a(0)\end{bmatrix} = \begin{bmatrix}x\\0\end{bmatrix}, \tag{19}$$

where $a(t) \in \mathbb{R}^p$ denotes a point in the augmented part, and $h(t) \in \mathbb{R}^d$ is the hidden state at time $t$.

Specifically, Augmented Neural ODEs can track back the ODE flow to recover the original input $x$ by using the hidden state $h(t)$ and the time information $t$. The input $x$ is the injection of the dynamics in HomoODE. The augmented part $a(t)$ can be viewed as the intermediate $z(t)$ in HomoODE. In Augmented Neural ODEs, we can treat the recovered input $x$ as the *condition* in HomoODE, improving its representation ability. Hence, Augmented Neural ODEs outperform Neural ODEs. However, the origin input $x$ computed by Augmented Neural ODEs may not be accurate enough. This probably is the reason why the performance of Augmented Neural ODEs is not competitive to HomoODE.

## 8 Experiments

To confirm the efficiency of HomoODE, we conduct experiments on several classical image classification datasets to compare our model with the previous implicit model, including DEQ [6], monDEQ [46], Neural ODE [10] and Augmented Neural ODE [12]. For ODE datasets, DEQs inherently are not designed to handle simulation tasks of continuous physical processes as DEQs do not involve the continuous time series in the models. Sequential datasets are not suitable as well due to the inconsistent backbones. For sequential datasets, DEQ models are based on transformer, while Neural ODEs are based on fully-connected layers. Therefore, we consider the image classification task is suitable for the fair comparison of HomoODE and baselines. Concretely, we evaluate the stability of the training process via the learning curve of accuracy in the test dataset and exhibit the performance of different implicit models in terms of accuracy, memory consumption, and inference time. It is worth noting that we also perform HomoODE with & without the data augmentation and the adjoint backpropagation technique to check their impacts on our model. Besides, we also contrast HomoODE with zero vector initialization and learnable initialization to assess the capability of our acceleration method. Our code is available at: `https://github.com/wadx2019/homoode`.

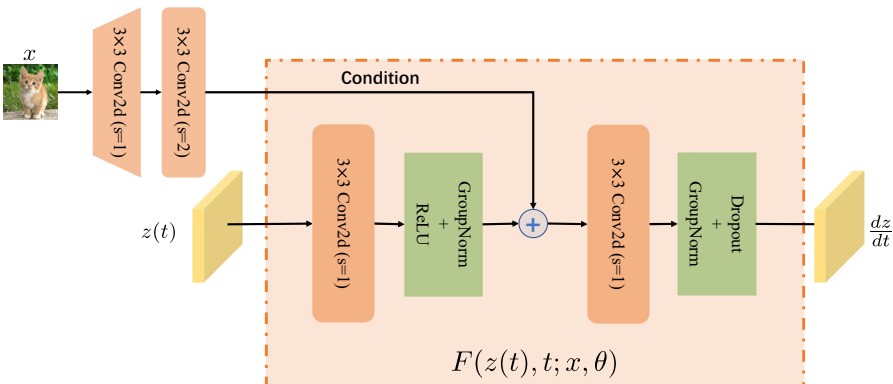

Figure 3: The deployed neural network architecture in HomoODE. Here, $s$ denotes the stride and the channel number of all convolutional layers is $64$.

**Experimental setup.** HomoODE is performed on several standard datasets, including CIFAR-10 [34], SVHN [38], and MNIST [35]. As shown in Figure 3, HomoODE contains several simple convolutional layers. Its network structure is not specially designed for image classification tasks like MDEQ [7]. Notably, the memory consumption of HomoODE is less than that of other implicit models as reported in [46, 48, 7]. As we discussed in Section 5, the time information $t$ is not fused into the input of HomoODE, unlike Neural ODE. Besides, we optimize the shared initial information

$\tilde{z}_0$ using SGD optimizer with the learning rate 0.02 and perform the update once every 20 updates for HomoODE.

**Comparison with former implicit models.** Table 1 presents the performance of HomoODE with different settings and other implicit models in the CIFAR-10 dataset. It can be observed that HomoODE outperforms the previous implicit models in terms of both accuracy and memory consumption. Moreover, the inference time of HomoODE is much faster than DEQ and its variants. Notably, we also find that DEQ runs slower with larger test batch sizes. This may be due to the inefficiency of DEQ for parallel computation with large test batch sizes. In contrast, HomoODE does not exhibit this problem. Additionally, there is a large increase in accuracy in our model when the data augmentation technique is applied. It suggests that HomoODE learns better and more complex representations through augmented datasets. Furthermore, given that we adopt the dopri5 ode solver with an adaptive step, this complex representation may result in increased NFE and inference time. Overall, these phenomenons validate that Ho-

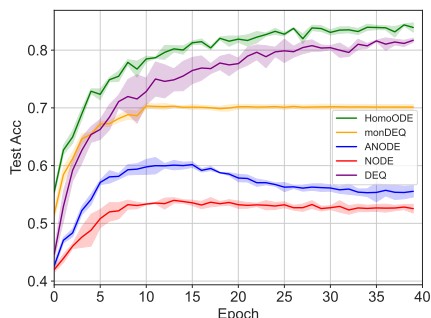

Figure 4: Learning curve of different implicit models on CIFAR-10 datasets across 5 runs without data augmentation. The x-axis denotes the epoch during training and the y-axis denotes the accuracy of models on the test datasets.

moODE has a more powerful representation ability compared to other implicit models with similar model capacity. Extensive experiments in SVHN and MNIST also confirm these properties of our model as shown in Table 2.

Besides, we also plot the learning curves of different algorithms in Figure 4. The results demonstrate the stability of training HomoODE compared with other methods and exhibit that HomoODE is not prone to over-fitting, whereas other ODE-based models may suffer from that.

**Efficiency of the learnable initialization.** Figure 5 illustrates that the learnable initialization trick can improve HomoODE with about $2.5\times$ speedup in the inference time than before. This impact is obvious in both cases with & without the adjoint backpropagation technique. In addition, the corresponding test accuracy during the training process also reflects this acceleration technique does not bring a loss in the performance of our model. Surprisingly, it even brings a slight improvement in the case of the adjoint backpropagation technique. This is probably because a good initial point can decrease the total length of zero path $s$, which reduces the gradient error induced by using the adjoint method.

## 9 Conclusion

In this paper, we show that both DEQs and Neural ODEs can be viewed as a procedure for solving an equilibrium-point-finding problem via the theory of homotopy continuation. Motivated by this observation, we illustrate that these two implicit models are actually the two sides of the same coin. Specifically, DEQs inject the input information as the *condition* into the equilibrium-point-finding problem $z^\star = f(z^\star; x, \theta)$ while Neural ODEs fuse the input information into the initial point. Further, we propose a novel implicit model called HomoODE, which inherits the advantages of both DEQs and Neural ODEs. Our experiments indeed verify that HomoODE outperforms both DEQs and Neural ODEs while avoiding the instability of the training process, that is often observed with DEQs. Moreover, we developed a method to speed up HomoODE and the ODE-solving operation by almost three times by using a shared learnable initial point. Overall, the experimental results on several classical image classification datasets demonstrate the efficiency of HomoODE in terms of both accuracy and memory consumption.

Although this paper offers a brand new perspective on implicit models, we also want to highlight a limitation of this idea, Actually, we do not present an explicit form of the equilibrium transformation function, which is implicitly determined by a modified neural ODE. Besides, while HomoODE has a powerful representation ability, the equilibrium-point-solving procedure of it is implicit, which

| Method | Model size | Inference Time | Accuracy |
|---|---|---|---|
| DEQ [6] | 170K | $5.8\times$ | $82.2 \pm 0.3\%$ |
| monDEQ [46] | 172K | $1.6\times$ | $74.0 \pm 0.1\%$ |
| Neural ODE [10] | 172K | $3.2\times$ | $55.3 \pm 0.3\%$ |
| Aug. Neural ODE [12] | 172K | $1.7\times$ | $58.9 \pm 2.8\%$ |
| **HomoODE** | **132K** | **$1.2\times$** | **$85.8 \pm 0.1\%$** |
| **HomoODE$^\dagger$** | **132K** | **$1.0\times$** | **$83.2 \pm 0.4\%$** |
| **HomoODE$^\star$** | **132K** | **$1.4\times$** | **$90.1 \pm 0.2\%$** |
| **HomoODE$^{\star\dagger}$** | **132K** | **$1.0\times$** | **$88.4 \pm 0.1\%$** |

Table 1: Performance of HomoODE compared to previous implicit models on CIFAR-10. $^\star$ with data augmentation; $^\dagger$ with adjoint method. The inference time is expressed as a multiple of the inference time of HomoODE with the adjoint method and the inference batch size is $400$. Each result is obtained with 5 random runs.

| Dataset | Method | Model size | Accuracy |
|---|---|---|---|
| SVHN | DEQ [6] | 170K | $93.6 \pm 0.5\%$ |
| | monDEQ [46] | 170K | $92.4 \pm 0.1\%$ |
| | Neural ODE [10] | 172K | $81.0 \pm 0.6\%$ |
| | Aug. Neural ODE [12] | 172K | $83.5 \pm 0.5\%$ |
| | **HomoODE** | **132K** | **$95.9 \pm 0.1\%$** |
| MNIST | DEQ [6] | 80K | $99.5 \pm 0.1\%$ |
| | monDEQ [46] | 84K | $99.1 \pm 0.1\%$ |
| | Neural ODE [10] | 84K | $96.4 \pm 0.5\%$ |
| | Aug. Neural ODE [12] | 84K | $98.2 \pm 0.1\%$ |
| | **HomoODE** | **34K** | **$99.6 \pm 0.1\%$** |

Table 2: Performance of HomoODE compared to previous implicit models on SVHN and MNIST. Each result is obtained with 5 random runs.

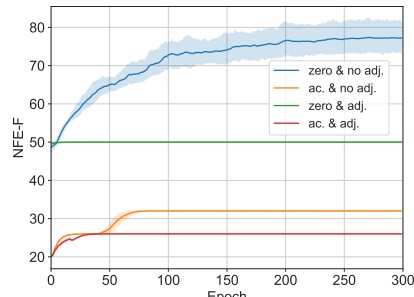

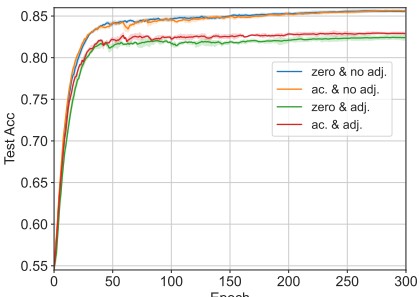

Figure 5: Learning curve of HomoODE with different settings on CIFAR-10 across 5 runs without data augmentation. The x-axis denotes the epoch during training. The y-axis (up) denotes the accuracy of models on the test dataset and the y-axis (down) denotes the NFE of them in the forward pass.

weakens its interpretability. Hence, exploring a more interpretable approach for the forward pass and backpropagation of HomoODE is under consideration in our future work.

## Acknowledgement

This work was supported by NSFC (No.62303319), Shanghai Sailing Program (22YF1428800, 21YF1429400), Shanghai Local College Capacity Building Program (23010503100), Shanghai Frontiers Science Center of Human-centered Artificial Intelligence (ShangHAI), MoE Key Laboratory of Intelligent Perception and Human-Machine Collaboration (ShanghaiTech University), and Shanghai Engineering Research Center of Intelligent Vision and Imaging.

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

# A  Background of Homotopy Continuation

Homotopy mapping $H(z, \lambda) = \lambda r(z) + (1 - \lambda)g(z)$ provides a continuous transformation by gradually deforming $g(z)$ into $r(z)$ while $\lambda$ increases from 0 to 1 in small increments. The solution to $r(z)$ can be found by following the zero path of the homotopy mapping $H(z, \lambda) = 0$. Usually, one can choose an artificial function $g(z)$ with an easy solution. Figure 6 shows the transformation of homotopy mapping with $\lambda$ increasing from 0 to 1, the homotopy function goes from an artificial, "easy" problem to the nonlinear problem in which we are interested.

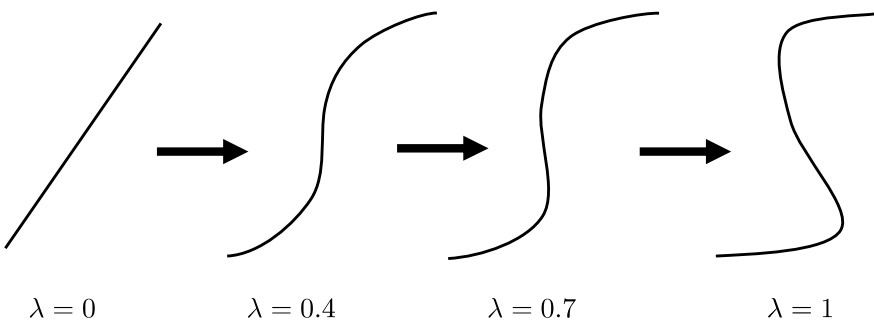

$$\lambda = 0 \qquad \lambda = 0.4 \qquad \lambda = 0.7 \qquad \lambda = 1$$

Figure 6: Transformation of homotopy mapping along with $\lambda$.

## A.1  Global Convergence of Homotopy Continuation

Here, we briefly recall the theoretical foundation of homotopy methods to show its global convergence with probability one.

**Definition 2** *Let $E^n$ denote $n$-dimensional real Euclidean space, let $U \subset E^n$ and $V \subset E^m$ be open sets, and let $H : U \times V \times [0, 1) \to E^n$ be a $\mathcal{C}^2$ mapping. $H$ is said to be transversal to zero if the Jacobian matrix $\nabla H$ has full rank on $H^{-1}(0)$.*

**Theorem 1.** *(Parametrized Sard's Theorem [1]) If $H(z_0, z, \lambda)$ is transversal to zero, then for almost all $z_0 \in U$ the mapping*

$$H_{z_0}(z, \lambda) = H(z_0, z, \lambda), \tag{20}$$

*is also transversal to zero; i.e., with probability one the Jacobian matrix $\nabla_{H_{z_0}}(\lambda, z)$ has full rank on $H_{z_0}^{-1}(0)$.*

The method for constructing a homotopy algorithm to solve the nonlinear system $r(z) = 0$ with global convergence is as follows: 1) $H(z_0, z, \lambda)$ is transversal to zero; 2) $H_{z_0}(z, 0) = H(z_0, z, 0)$ is trivial to solve and has a unique solution $z_0$; 3) $H_{z_0}(z, 1) = r(z)$; 4) $H_{z_0}^{-1}(0)$ is bounded.

Then for almost all $z_0 \in U$ there exist a zero path $s$ of $H_{z_0}$, along with the Jacobian matrix $\nabla H_{z_0}$ has rank $n$. The zero path starts from $(0, z_0)$ and reaching $z^\star$ at $\lambda = 1$. This zero path $s$ does not intersect itself, and it is disjoint from any other zero paths of $H_{z_0}$. Furthermore, if $\nabla r(z)$ is nonsingular, then the zero path $s$ has a finite arc length.

## A.2  Fixed Point Homotopy Continuation

One commonly used homotopy function to find solutions of $r(z) = 0$ is the Fixed Point Homotopy [11] given by:

$$H(z, \lambda) = \lambda r(z) + (1 - \lambda)(z - z_0), \tag{21}$$

where $z_0 \in \mathbb{R}^n$ and $\lambda$ in unit interval $[0, 1]$. At $\lambda = 0$, the starting system is $H(z, 0) = z - z_0 = 0$ for which the only solution is $z = z_0$. At $\lambda = 1$, the system $H(z, 1) = r(z) = 0$ is the system of equations of interest.

For $U \subset \mathbb{R}^n$, we use int $U$ to denote the interior of $U$. And we say $H$ is boundary-free at $\lambda_0 \in [0, 1]$ if $z \notin \partial U$ for any $z \in H|_{\lambda = \lambda_0}^{-1}(\{0\})$. Generally, we say $H$ is boundary-free for $\lambda$ in a subset $S \subset [0, 1]$ if $H$ is boundary-free for all $\lambda \in S$. The following theorem provides the fixed point of $f(z)$ under the existence of Fixed Point Homotopy.

**Theorem 2.** *(Fixed Point Theorem [11]) Given smooth function $f : U \to \mathbb{R}^n$, let $U \in \mathbb{R}^n$ be compact and $\text{int}U \neq \oslash$. For some $z_0 \in \text{int}U$, if $H : U \times [0,1] \to \mathbb{R}^n$ is boundary-free for $0 \leq \lambda \leq 1$, where*

$$H(z, \lambda) = \lambda(z - f(z)) + (1 - \lambda)(z - z_0), \tag{22}$$

*then f has a fixed point, i.e., there exists an $z^\star \in U$ such that $f(z^\star) = z^\star$.*

### A.3  Newton Homotopy

Another commonly used homotopy function is the Newton homotopy [11], which is defined as follows:

$$\begin{aligned} H(z, \lambda) &= \lambda r(z) + (1 - \lambda)[r(z) - r(z_0)] \\ &= r(z) - (1 - \lambda)r(z_0), \end{aligned} \tag{23}$$

where $r : \mathbb{R}^n \to \mathbb{R}^n$ is the smooth system of interest, and $z_0$ is a generically chosen point in $\mathbb{R}^n$.

Notably, there is a close connection between the Newton homotopy and the well-known Newton's method [2] for solving nonlinear equations. Given $\nabla r(z)$ is nonsingular, we can apply the differentiation on the zero path of the homotopy mapping, i.e. $H(z, \lambda) = 0$, yielding the initial value problem:

$$\frac{dz}{d\lambda} = -(\nabla r(z(\lambda)))^{-1} r(z_0), \tag{24}$$
$$z(0) = z_0.$$

Applying Euler's method at $\lambda = 0$ with step size 1 to the above ODE (24) from the initial point $z = z(0)$, the approximation of next iteration $z(1)$ becomes:

$$z(1) = z(0) - (\nabla r(z(0)))^{-1} r(z(0)). \tag{25}$$

Apparently, (25) is a single iteration of Newton's method. Hence, Newton's iteration can be considered as the application of Euler's method with step size 1 on the solution curve given by the Newton homotopy. However, in contrast to Newton's method, which is generally a local method, the Newton homotopy exhibits certain global convergence properties.

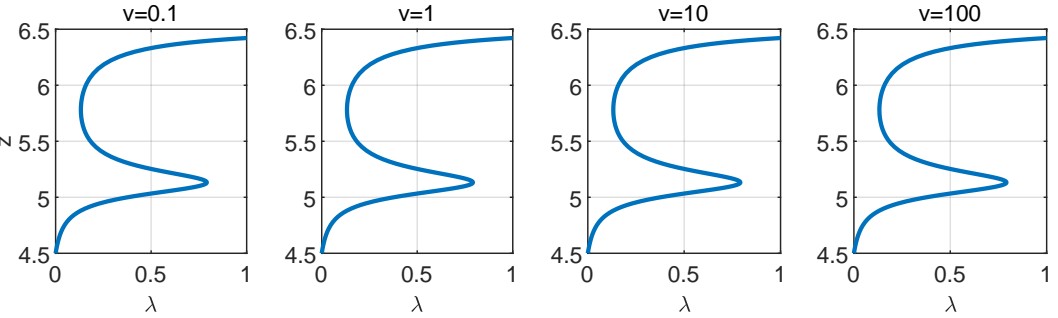

Figure 7: Iterative curve of homotopy continuation under different $v$.

## B  The impact of different values of velocity $v$ for the homotopy continuation

In Section 4 we introduce $v := \frac{ds}{dt}$ as the velocity of the point $(z, \lambda)$ traveling along the zero path, and modify the normalization condition by introducing $v$ into (3). Here we demonstrate that the convergence of the homotopy method is not affected by such modification by numerical experiments.

We choose a nontrivial nonlinear equation and solve it via homotopy continuation under different velocities $v$. Figure 7 shows the zero paths of homotopy continuation under different velocities $v$. Apparently, the zero paths are all on the same trajectory and heading for the same solution. Hence, the convergence of the homotopy method doesn't depend on the change in the value of velocity. With the modified normalization condition, the underlying dynamics become:

$$\frac{dz}{dt} = (I - \lambda(z)\nabla_z f(z; \theta))^{-1} f(z; \theta) \sqrt{v^2 - \left\| \frac{dz}{dt} \right\|^2}. \tag{26}$$

Following Neural ODEs, we can employ neural networks to approximate the differential equation above. The existence of a corresponding equilibrium-point-finding problem for Neural ODE is also guaranteed by the modified normalization condition. Specifically, one can refer to (11) (12). Hence, Neural ODEs can be regarded as the procedure of solving an equilibrium-point-finding problem with homotopy continuation.

## C  Bridging DEQ and Neural ODE from another type of Homotopy method

We have established the connection between DEQs and Neural ODEs via Fixed Point Homotopy in Section 4. Here, we also show that we can show a similar connection using another type of Homotopy method, i.e., Newton Homotopy.

By defining $r(z) = z - f(z; \theta)$, and setting $z_0 = \mathbf{0}$ for simplification, from (23) we have:

$$H(z, \lambda) = z - f(z; \theta) + (1 - \lambda)f(0; \theta), \tag{27}$$

Different from Fixed Point Homotopy, we can associate $z$ directly with $\lambda$ without introducing $v$. From (24) we have:

$$\frac{dz}{d\lambda} = (I - \nabla_z f(z(\lambda); \theta))^{-1} f(z(\lambda); \theta), \quad z(0) = \mathbf{0}. \tag{28}$$

Here we can view $\lambda$ as the time information for the differential equation, and $z(1)$ is the solution $z^\star$ of the corresponding equilibrium-point-finding problem when $\lambda = 1$. We can also employ neural networks to approximate the differential equation as Neural ODEs do. Therefore, Neural ODEs can also be regarded as the procedure of solving an equilibrium-point-finding problem via Newton Homotopy.

## D  Stability of HomoODE

This section demonstrates the stability of HomoODE based on Picard–Lindeöf Theorem [29].

**Theorem 3.** *(Picard–Lindeöf Theorem [29]) Let $I : [a, b]$ be an interval, let $f : I \times \mathbb{R}^n \to \mathbb{R}^n$ be a function, and let*

$$z'(t) = f(t, z(t)), \tag{29}$$

*be the associated ordinary differential equation. If $f$ is Lipschitz continuous in the second argument $z$, then this ODE possesses a unique solution on $[a, a+\epsilon]$ for each possible initial value $z(0) = z_0 \in \mathbb{R}^n$, where $\epsilon < \frac{1}{L}$, $L$ is the Lipschitz constant of the second argument of $f$.*

HomoODE solves the equilibrium-point-finding problem implicitly using a modified Neural ODE via homotopy continuation. Hence it also has the stability of Neural ODE, which can be explained by the Picard–Lindeöf Theorem. Assuming the underlying dynamic of HomoODE is Lipschitz continuous in $z$, then both existence and uniqueness can be guaranteed by the Picard–Lindeöf Theorem.

## E  Accelearation for HomoODE

In Section 6, it is mentioned that the update on the shared initial point $z_0$ is equivalent to maintaining a dynamic geometrical center of the equilibrium points of all the samples. We will show the correctness of this proposition and illustrate the relationship between the learning rate and the step of the dynamic update.

**Proposition 1.** *The update on the shared initial point $z_0$ is equivalent to maintaining a dynamic geometrical center of the equilibrium points of all the samples.*

*Proof.* Recall that the loss function of $z_0$ is defined as $\mathcal{L}(z_0) := \mathbb{E}_{x \sim \mathcal{D}} \left[ (z^\star(x) - z_0)^2 \right]$. According to the definition of the variance, we have

$$\mathrm{Var}(z^\star(x) - z_0) = \mathbb{E} \left[ (z^\star(x) - z_0)^2 \right] - \mathbb{E} \left[ z^\star(x) - z_0 \right]^2. \tag{30}$$

Define $\bar{z}^\star := \mathbb{E}(z^\star(x))$, we obtain

$$
\begin{aligned}
\mathbb{E}\left[(z^\star(x) - z_0)^2\right] &= \mathrm{Var}\left(z^\star(x) - z_0\right) + \mathbb{E}\left[z^\star(x) - z_0\right]^2 \\
&= \mathrm{Var}\left(z^\star(x) - z_0\right) + (\bar{z}^\star - z_0)^2 \\
&= \mathrm{Var}\left(z^\star(x)\right) + (\bar{z}^\star - z_0)^2
\end{aligned}
\tag{31}
$$

Here $\mathrm{Var}\left(z^\star(x)\right)$ is irrelevant to $z_0$. Hence, $\min_{z_0} \mathbb{E}\left[(z^\star(x) - z_0)^2\right]$ is equivalent to minimizing $\min_{z_0} (\bar{z}^\star - z_0)^2$, which means the update on $z_0$ is equivalent to maintaining a geometrical center of the equilibrium points of all the samples. $\square$

According to Proposition 1, we can also write the update as the form of $z_0 = \alpha \bar{z}^\star + (1-\alpha)z_0$, which is more straightforward. Therefore, we also want to further explore the relationship between the learning rate $\eta_{\text{init}}$ and update step $\alpha$. Consider the gradient descent on $z_0$

$$
\begin{aligned}
z_0 &= z_0 - \eta \nabla_{z_0} \mathbb{E}_{x \sim \mathcal{D}}\left[(z^\star(x) - z_0)^2\right] \\
&= z_0 + 2\eta\left(\bar{z}^\star(x) - z_0\right) \\
&= 2\eta \bar{z}^\star(x) + (1 - 2\eta)z_0.
\end{aligned}
\tag{32}
$$

Since $z_0$ is the broadcast tensor from initial information $\tilde{z}_0$, the actual gradient descent has the form of

$$
\tilde{z}_0 = \frac{2}{hw}\eta_{\text{init}}\bar{z}^\star(x) + (1 - \frac{2}{hw}\eta_{\text{init}})\tilde{z}_0.
\tag{33}
$$

where $h, w$ are the height and width of the feature map respectively. Finally, we obtain the relationship between $\eta_{init}$ and $\alpha$ is $\eta_{\text{init}} = \frac{hw}{2}\alpha$. This means we can set a large learning rate for $\eta_{\text{init}}$ even greater than 1, especially when the feature map is large.

## F    Adjoint Method for HomoODE

---
**Algorithm 1** Adjoint Method for HomoODE
---
**Input:** initial point $z(t_0)$, *condition* $x$, parameter $\theta$, start time $t_0$, stop time $t_1$

$\quad s_0 = [z(t_1), \frac{\partial L}{\partial z(t_1)}, 0_x, 0_\theta]$ $\qquad\qquad\qquad$ ▷ Define initial augmented state

$\quad$ **def** aug_dynamics$(([z(t), a(t), \cdot, \cdot], t, \theta))$: $\qquad$ ▷ Define dynamics on augmented state

$\quad\quad$ **return** $\left[F(z(t), t; x, \theta); -a(t)^\top \frac{\partial f}{\partial z}, -a(t)^\top \frac{\partial f}{\partial x}, -a(t)^\top \frac{\partial f}{\partial \theta}\right]$ ▷ Compute vector-Jacobian products

$\quad [z(t_0), \frac{\partial L}{\partial z(t_0)}, \frac{\partial L}{\partial x}, \frac{\partial L}{\partial \theta}] = \text{ODESolve}(s_0; \text{aug\_dynamics}; t_1; t_0; \theta)$ $\quad$ ▷ Solve reverse-time ODE

**return** $\frac{\partial L}{\partial x}, \frac{\partial L}{\partial \theta}$ $\qquad\qquad\qquad\qquad\qquad$ ▷ Return gradients with respect to $x$ and $\theta$
---

The adjoint method [10, 42] is an efficient backpropagation method that can save the memory footprint in Neural ODE during training. However, there is a minor difference when we apply the adjoint method in the training of HomoODE. Unlike Neural ODE which computes the gradient with respect to $z(t_0)$, our HomoODE calculates the gradient with respect to the *condition* $x$ instead. Accordingly, we modify the adjoint method in Neural ODE and present the computation procedure in Algorithm 1.

## G    Additional Experimental Results

We conducted additional experiments in CIFAR-100 [34] and Tiny ImageNet [37] to validate the potential of applying HomoODE to difficult image classification tasks with larger model sizes. Specifically, we extend the channel numbers of the convolutional layers to 128 in HomoODE and increase the model size of the compared implicit models correspondingly for fairness. The experiments in CIFAR-100 and Tiny ImageNet are all implemented with data augmentation. The concrete operations in data augmentation involve zero-padding the 32×32 images to 40×40 and then performing random horizontal flips. As shown in Table 4, the performance of HomoODE is much better than other implicit models in terms of both memory consumption and test accuracy.

| Method | Model size | Accuracy |
|--------|-----------|----------|
| DEQ [6] | 770K | $64.5 \pm 0.7\%$ |
| monDEQ [46] | 1M | $59.8 \pm 0.3\%$ |
| Neural ODE [10] | 874K | $31.7 \pm 0.6\%$ |
| Aug. Neural ODE [12] | 857K | $36.2 \pm 0.9\%$ |
| **HomoODE**$^\dagger$ | **565K** | **$69.30 \pm 0.1\%$** |
| **HomoODE** | **565K** | **$71.57 \pm 0.2\%$** |

Table 3: Additional experiments on CIFAR-100. HomoODE$^\dagger$ and HomoODE represent the training with and without the adjoint method, respectively. The experiments are all implemented with data augmentation.

| Method | Model size | Accuracy |
|--------|-----------|----------|
| DEQ [6] | 810K | $47.6 \pm 0.4\%$ |
| monDEQ [46] | 783K | $24.83 \pm 0.1\%$ |
| **HomoODE**$^\dagger$ | **617K** | **$54.14 \pm 0.2\%$** |
| **HomoODE** | **617K** | **$56.1 \pm 0.1\%$** |

Table 4: Additional experiments on Tiny-ImageNet. HomoODE$^\dagger$ and HomoODE represent the training with and without the adjoint method, respectively. The experiments are all implemented with data augmentation.

Besides, we also performed the ablation study on the learning rate for the shared initial point as shown in Figure 8. Notably, the ablation study is implemented with the adjoint method. Given the practicality of the adjoint method in handling the backpropagation of large deep models, we are inclined to explore a wider range of characters when utilizing it. It can be observed that HomoODE is robust to the learning rate for the shared initial point and there exists a slight improvement when $\mathrm{lr} = 0.1$. We believe the reason is the same as the slight improvement in contrast experiments of HomoODE with/without acceleration. That is, the larger learning rate for the shared initial point makes the initial point $z_0$ closer to the current geometrical center of the equilibrium points of all samples, which further leads to the shorter zero path that the adjoint method goes through. Finally, it reduces the error of backpropagated gradient.

## H   Hyper-parameter Settings

Our experiments are implemented on a GPU of NVIDIA GeForce RTX 3090 with 24GB. The hyper-parameters applied in HomoODE are shown in Table 5. Experiments related to other implicit

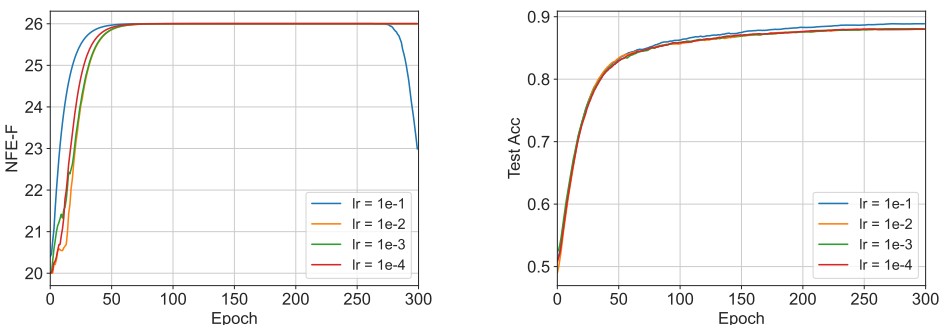

Figure 8: Ablation study on the learning rate for the shared initial point.

models are based on deq [2], monotone_op_net [3] and augmented-neural-odes [4]. Notably, the dropout layer in our experiments follows the variational dropout operation in [7] because the traditional dropout operation hurts the stability of convergence to the equilibrium.

| Parameter | MNIST | SVHN | CIFAR-10 | CIFAR-100 |
|---|---|---|---|---|
| Batch Size | 64 | 64 | 64 | 64 |
| Optimizer | Adam | Adam | Adam | Adam |
| Learning Rate | 0.001 | 0.001 | 0.001 | 0.001 |
| Frequency of Initial Point Update | 20 | 20 | 20 | 5 |
| Optimizer for Initial Point | SGD | SGD | SGD | SGD |
| Learning Rate for Initial Point | 0.02 | 0.02 | 0.02 | 0.01 |
| Variational Dropout Rate | 0.1 | 0.1 | 0.1 | 0.15 |
| Number of Channels | 32 | 64 | 64 | 128 |
| Absolute Tolerance for ODE Solver | 1E-3 | 1E-3 | 1E-3 | 1E-3 |
| Relative Tolerance for ODE Solver | 1E-3 | 1E-3 | 1E-3 | 1E-3 |

Table 5: Hyper-parameters used in HomoODE under different image classification tasks.

# I   Details on Tested Equation in Section 6

With respect to the details on the tested nonlinear equation mentioned in Section 6, we randomly choose an equation with high nonlinearity that is

$$f(x) = 2x + \exp(-0.1x) + 5\sin(4x) - 16 = 0,$$

and one solution of it is $6.4217$. Our experiment shown in Figure 2 is based on the initial points around this solution. Actually, we found that even if we modify the parameters in this function arbitrarily, a similar result can be obtained as well.

# J   More Discussions on Learnable Initial Point

It is worth noting that we also carried out some trials with an input-based initial point predictor [8] to accelerate HomoODE. However, the performance of the initial point predictor in HomoODE is not desirable in terms of both inference time and test accuracy. Based on the comprehensive analysis we provided in Section B, C, this is probably because too frequent or too large updates on the initial point will destroy the stable link between ODE function $F(z(t), t; x, \theta)$ and the equilibrium transformation function $f(z; x, \theta)$. In particular, the additionally-introduced variable $v$ will change sharply when a large change is applied to the initial point. In this case, the function $\lambda(t)$ will change dramatically, eventually ruining the trained equilibrium transformation function $f(z; x, \theta)$. This phenomenon can also be illustrated from the perspective of Newton homotopy. Note that the ODE function is actually determined by the initial point. Therefore, once the initial point is changed, the ODE function will change and then the trained ODE function will be ruined.

---

[2] https://github.com/locuslab/deq
[3] https://github.com/locuslab/monotone_op_net
[4] https://github.com/EmilienDupont/augmented-neural-odes

