# OpenReview forum: "Two Sides of The Same Coin: Bridging Deep Equilibrium Models and Neural ODEs via Homotopy Continuation"
_NeurIPS.cc/2023/Conference — NeurIPS 2023 poster_

### Official Review · Reviewer_ZDnx · 2023-07-06

**Soundness:** 3 good
**Presentation:** 3 good
**Contribution:** 3 good
**Rating:** 6
**Confidence:** 4

**Summary:**

The paper shows that the fixed point equation solved by a Deep Equilibrium Model (DEQ) can be written as a Neural ODE.This is achieved by first introducing the equation of the solution path for the Homotopy equation, which is referred to as the Fixed Point Homotopy equation, and then deriving the appropriate homotopy mapping for the fixed point equation solved by a DEQ. Hence, they introduce an ODE that solves for a fixed point of a system (using homotopy continuation) and therefore claim that DEQs and neural ODEs are “essentially two-sides of the same coin”.

The authors then introduced HomoODE, that uses the input injection from a DEQ and the derived fixed-point-ODE to solve for a task, however instead of solving for the fixed point using newton-method in a DEQ homoODE uses a normal ode-solver similar to neural ODEs. The input injection helps in getting better performance than neural ODE and the implicit solving using ODE solver results in a more memory efficient methodology

**Strengths:**

The paper is well written, and effectively establishes the connection between neural ODEs and DEQs. Furthermore, they introduce HomoODE, which inherits the better performance capabilities of a DEQ, and has a lesser memory footprint (thanks to the ODEsolver instead of newton method based solver, used in Neural ODEs).

The experiments show on various classification tasks that HomoODE matches or beats DEQs in various benchmarks with fewer model parameters and with faster inference time.

**Weaknesses:**

There are some details missing in some parts of the paper, I have pointed them out in the Questions section.

**Questions:**

- In lines 205-210 the authors mention that neural ODE is a fixed-point problem with a “fixed-condition” x with different initial point z(t_0). I am not sure if I get that point. Presumably the fixed point used to derived the fixed point ODE could be conditioned on the input $x$ as well.
- The derived equation in equation 12 has $F(z(t), t; x; \theta)$ under the square root, is that a typo? or, there is some kind of “input injection” of $x$ that is implicitly assumed in Neural ODE that solves an equilibrium-point-finding problem.
- How is $v$, the velocity in equation 9 chosen in the final neural ODE?
- In Section 6, the authors mention the homotopy continuation on a nonlinear equation in using matlab. The details on which equation and what was the purpose of that experiment is missing. I am unsure about the conclusion of that section, do that authors learn a common initialization for all samples $x$ in the dataset, and is that learned or just an average over the entire dataset?

**Limitations:**

Authors have discussed the limitations of the work.

---

> ### Author Rebuttal · Authors · 2023-08-09
>
> Thanks for your valuable comments and suggestions. Here we address your detailed comments as below:
>
> > **Q1**: In lines 205-210 the authors mention that neural ODE is a fixed-point problem with a “fixed-condition” $x$ with different initial point $z(t_0)$. I am not sure if I get that point. Presumably the fixed point used to derive the fixed point ODE could be conditioned on the input $x$ as well.
>
> **A1**: Thank you for pointing out the lack of clarity here. There is a typo here and it should be "Neural ODEs solve the fixed equilibrium-point-finding problem with different initial points $z(t_0)$". There is no condition in Neural ODEs and here we want to state the underlying equilibrium-point-finding problem in Neural ODEs does not vary with the input. We will modify this statement in the final version.
>
> > **Q2**: The derived equation in equation 12 has $F(z(t),t;x;\theta)$  under the square root, is that a typo? or, there is some kind of “input injection” of  $x$ that is implicitly assumed in Neural ODE that solves an equilibrium-point-finding problem.
>
> **A2**: Yes, it is a typo. Thank you for your careful reading. $F(z(t),t;x;\theta)$ should be $F(z(t),t;\theta)$. We will fix this issue in the final version.
>
> > **Q3**: How is $v$, the velocity in equation 9 chosen in the final neural ODE?
>
> **A3**: We would like to clarify that $v$ is an auxiliary variable for the theoretical analysis to bridge the DEQs and Neural ODEs via Fixed Point Homotopy (line 178-184). In the experiments, we do not need to include $v$ as a hyperparameter or compute its value because it has been implicitly contained in the neural network.
>
> > **Q4(1)**: In Section 6, the authors mention the homotopy continuation on a nonlinear equation in using matlab. The details on which equation and what was the purpose of that experiment is missing.
>
>
> **A4(1)**: Thank you for raising the concern. As illustrated in line 257-261, we would like to explain via Figure 2 that it is **not necessary** to train the initial points for its corresponding image very accurately. This is because homotopy continuation is different from methods like Newton's method where even if we set a initial point very close to the solution, it will still need **certain iteration times** to obtain the solution. That's why we only maintain a common initial point for all samples $x$ rather than calculate a sample-conditioned initial point like [8].
>
> With respect to the detail on the tested nonlinear equation, we randomly choose an equation with high nonlinearity that is
> $$
> f(x) = 2x+\exp(-0.1x)+5\sin(4x)-16 = 0,
> $$
> and one of the solutions is 6.4217. Our experiments in homotopy continuation are based on the initial points around this solution. Actually, you can modify the parameters in this function arbitrarily and can obtain similar conclusion as well. We will add these details in the final version.
>
> > **Q4(2)**: I am unsure about the conclusion of that section, do that authors learn a common initialization for all samples $x$ in the dataset, and is that learned or just an average over the entire dataset?
>
> **A4(2)**: Yes, we learn a common initialization for all samples $x$ in the dataset. With the training process, the fixed-point solutions of all samples are changing as well. To ensure a stable training process, the common initialization is learned with stochastic gradient descent instead of directly averaging among the sample solutions.

---

> > ### Comment · Reviewer_ZDnx · 2023-08-17
> >
> > Thank you for the reply and adding the clarification on the details for the nonlinear equation and the initialization! I understand the details of the methodology much better now!

---

### Official Review · Reviewer_iBob · 2023-07-06

**Soundness:** 4 excellent
**Presentation:** 4 excellent
**Contribution:** 4 excellent
**Rating:** 8
**Confidence:** 4

**Summary:**

This paper unifies DEQs and Neural ODEs via homotopy continuation. The authors propose a novel implicit model, HomoODE, which inherits the property of high accuracy from DEQs and the property of stability from Neural ODEs. Moreover, they develop an acceleration method by borrowing the idea from the DEQ-solver.

**Strengths:**

1 This paper reveals the inherent connection between two typical implicit models, i.e., DEQs and ODEs.  This connection is really insightful. This paper, in my opinion, contributes significantly to our knowledge of implicit models.

2 The proposed HomoODE combines the advantages of DEQs and ODEs. The authors leverage homotopy continuation, and thus HomoODEs do not suffer from the issue of unique equilibrium like DEQs. The proposed acceleration method is also nice. The authors provide a new design paradigm for implicit models.



**Weaknesses:**

1 This paper argures that "DEQs and Neural ODEs are two sides of the same coin", but compared with the analysis on  "two sides" is less than that on "the same coin."

2 More experiments on large scale datasets, e.g., ImageNet, would be nice.

**Questions:**

Please see weaknesses.

**Limitations:**

The authors have adequately addressed the limitations, and there is no potential negative societal impact of their work.

---

> ### Author Rebuttal · Authors · 2023-08-09
>
> We gratefully appreciate the reviewer for recommending acceptance of our submission and thank his/her valuable suggestions to help improve it. We will answer all the questions that the reviewers concern about.
>
> > **Q1**: This paper argues that "DEQs and Neural ODEs are two sides of the same coin", but compared with the analysis on "two sides" is less than that on "the same coin."
>
> **A1**: Thanks for pointing out the lack of more detail about the "two sides".
>
> DEQs involve an underlying equilibrium-point-finding problem, which varies on different conditions $x$. As a result, DEQs can map inputs to diverse representations and thus enjoys good performance. However, DEQs suffer from stability issues since the existence of unique fixed points cannot be guaranteed.
>
> Unlike DEQs, the equilibrium-point-finding problem in Neural ODEs does not vary with the initial points $z(t_0)$. The structure of the Neural ODEs can guarantee stability, but the same underlying problems will limit its representation capability. Therefore, we claim that DEQs and Neural ODEs are actually two sides of the same coin.
>
> We will add more analyses in the camera-ready version.
>
> > **Q2**: More experiments on large-scale datasets, e.g., ImageNet, would be nice.
>
> **A2**: Thanks for your valuable suggestion. Due to time limitations, our experiments on ImageNet are still in progress. However, to confirm the capability of our model in larger and more difficult datasets, we conducted the experiments on Tiny-ImageNet, which include a subset of the ImageNet dataset. Here we show the results of HomoODE compared with other implicit models.
>
> |  Method  | Model Size  |    Accuracy     |
> |:------------------:|:-----------:|:---------------:|
> | DEQ                | 810k        | 47.60 ± 0.4%    |
> | monDEQ             | 783k        | 24.83 ± 0.1%    |
> | **HomoODE**        | 617k        | 56.1 ± 0.1%     |
> | **HomoODE (adj.)** | 617k        | 54.14 ± 0.2%    |
>
>
> Moreover, the results on CIFAR-100 presented in Table 3 of the Appendix can also demonstrate certain capabilities of HomoODE on large-scale datasets.

---

### Official Review · Reviewer_aBa7 · 2023-07-06

**Soundness:** 3 good
**Presentation:** 3 good
**Contribution:** 2 fair
**Rating:** 5
**Confidence:** 4

**Summary:**

The paper uses homotopy continuation to show that Deep Equilibrium Models and Neural ODEs are effectively equivalent. They use the best properties of both worlds: DEQs for their higher accuracy and Neural ODEs for their stability to propose a new implicit method HomoODE.

**Strengths:**

* Theoretical explorations of the relationship between Neural ODEs and DEQs are limited, and the paper does a good job of linking the two ideas.

**Weaknesses:**

* Reformulating DEQs as Neural ODEs have been done in a prior work [1], which has not been mentioned or compared against
* MNIST and SVHN are too small-scale experiments and can lead to incorrect extrapolations on the method's benefits. I would recommend including larger problems like ImageNet.
* Some information on the baseline would be useful: Typically, solving discrete dynamical systems (DEQs) is faster than a continuous dynamical system. So the 8x inference time might be an artifact of poor solver choice for the DEQ.

[1] Pal, Avik, Alan Edelman, and Christopher Rackauckas. "Continuous Deep Equilibrium Models: Training Neural ODEs faster by integrating them to Infinity." arXiv preprint arXiv:2201.12240 (2022).

**Questions:**

See the section on weaknesses

---

> ### Author Rebuttal · Authors · 2023-08-09
>
> We thank the reviewer for your valuable feedback and constructive comments! We itemize the weaknesses or comments you mentioned and answer to them.
>
> > **Q1**: Reformulating DEQs as Neural ODEs have been done in a prior work [R4], which has not been mentioned or compared against
>
> **A1**: Thank you for your constructive suggestion. We have read this paper carefully and will cite it in our final version. However, we would like to point out that the goal and the studied problem in our work is different from those in [R4]. Our work attempts to establish an inherent connection between **general DEQ** and **general neural ODE** via homotopy continuation and then develop the intermediate model HomoODE to confirm our theory, while [R4] aims to reduce the computational expense of DEQ in training and inference via utilizing an **"infinity time" neural ODE**.
>
> In other words, our paper is for the theoretical analysis of the **inherent connection** rather than just reformulating DEQs or Neural ODEs to a special Neural ODE or DEQ to improve some aspects of their performance.
>
> Besides, based on the performance reported in [R4], HomoODE achieves better accuracy than Skip DEQ and Skip Reg. DEQ with even smaller model size. Here is the comparison in CIFAR-10 dataset.
>
> |   Method          |  Moel Size |   Accuracy    |
> |:-----------------:|:----------:|:-------------:|
> |   Skip DEQ [R4]    |  200K      |   82.0 ± 0.3% |
> | Skip Reg. DEQ [R4] |  164K      |   81.1 ± 0.3% |
> |**HomoODE (adj.)** |  132K      |  88.4 ± 0.1%  |
> |   **HomoODE**     |  132K      |   90.1 ± 0.2% |
>
>
> [R4] Pal, Avik, Alan Edelman, and Christopher Rackauckas. "Continuous Deep Equilibrium Models: Training Neural ODEs faster by integrating them to Infinity." arXiv preprint arXiv:2201.12240 (2022).
>
> > **Q2**: MNIST and SVHN are too small-scale experiments and can lead to incorrect extrapolations on the method's benefits. I would recommend including larger problems like ImageNet.
>
> **A2**: Apart from MNIST and SVHN, **we did conducted experiments on CIFAR-10 and CIFAR-100** (Please refer to Appendix G). All these experiments show the superiority of our HomoODE. Due to time limitations, the experiments on ImageNet are still in progress. **We chose the Tiny-ImageNet dataset, a subset of ImageNet, as supplementary experiments during the rebuttal phase. The results shown in the following table also verify the superiority of our model HomoODE.**
>
> The following table shows the result of experiments on Tiny-ImageNet.
> |  Method          | Model Size  |    Accuracy     |
> |:------------------:|:-----------:|:---------------:|
> | DEQ                | 810k        | 47.60 ± 0.4%    |
> | monDEQ             | 783k        | 24.83 ± 0.1%    |
> | **HomoODE**        | 617k        | 56.1 ± 0.1%     |
> | **HomoODE (adj.)** | 617k        | 54.14 ± 0.2%    |
>
>
> > **Q3 (1)**: Some information on the baseline would be useful: Typically, solving discrete dynamical systems (DEQs) is faster than a continuous dynamical system.
>
> **A3 (1)**: We respectively disagree with this comment. The iteration number required by solving discrete dynamical systems (DEQs) may be smaller than a continuous dynamical system, but the total running time may be still larger. This is because DEQs need nontrivial efforts in calculating the approximation of inverse Jacobian $B$ at each iteration, while this procedure is not needed in ODE-based models.
>
> In our experiments of DEQs, we followed the hyperparameter setting of DEQ on CIFAR-10 and chose Broyden solver for both forward and backward, and the threshold for the forward process is 18 and 20 for the backward process. Our experiments are implemented on a GPU of NVIDIA GeForce RTX 3090 with 24GB.
>
> **Q3 (2)**: The 8x inference time might be an artifact of poor solver choice for the DEQ.
>
> **A3 (2)**: Thank you for raising the concern. We have doubled checked the inference time in Table I, and there is a mistake as we ran multiple programs in parallel on the same GPU before. Now we have fixed this issue and updated the correct inference time in the following table. The new experiments are conducted with two different settings with test batch size 400 (used in our previous experiment) and test batch size 32 (newly added). We can see that DEQ still runs slow and even slower with large test batch sizes. This may be due to the inefficiency of DEQ for parallel computation with large test batch sizes.
>
> |  Method          | Inference Time (Batch size 400) | Inference Time (Batch size 32)|
> |:----:|:----:|:----:|
> | DEQ  | 5.8$\times$  | 2.2$\times$ |
> | **HomoODE**  |  1.2$\times$| 1.2$\times$ |
> | **HomoODE (aug.)**  | 1.4$\times$ | 1.4$\times$ |
> | **HomoODE (adj.)**  |  1.0$\times$  | 1.0$\times$ |
> | **HomoODE (adj.,aug.)**  |  1.0$\times$ | 1.0$\times$ |
> | monDEQ  | 1.6$\times$ | 1.7$\times$ |
> | NODE  | 3.2$\times$ | 1.8$\times$ |
> | ANODE  | 1.7$\times$ | 1.2$\times$ |

---

> > ### Comment · Reviewer_aBa7 · 2023-08-15
> >
> > **(A1)** Thanks for clarifying; I accept that the paper adds a valuable contribution in the form of a theoretical connection.
> >
> > **(A2)** Thanks for the additional experiments; however, I would still appreciate experiments on ImageNet (Proper) in the final version. But I acknowledge that given the short time, it is hard to produce those results during the rebuttal phase.
> >
> > **(A3)** I feel that is an unfair comparison. DEQs typically use Broyden based method simply because no other non linear solvers were ever compared. There are several better methods in terms of computational complexity for larger problems which can avoid inverting the jacobian, some examples would be JFNK and DFSane [1]. The reason no Jacobian inversion is needed in the ODE solvers used, is due the system being non-stiff. So it would be unfair to compare a nonlinear solver not designed for large scale problems against ODE solvers used for larger systems.
> >
> > Based on the rebuttal, I have updated my score to 5.
> >
> > [1] https://www.researchgate.net/publication/220576479_Spectral_Residual_Method_without_Gradient_Information_for_Solving_Large-Scale_Nonlinear_Systems_of_Equations/link/0fcfd5076febc07fb9000000/download

---

> > > ### Author Response · Authors · 2023-08-16
> > > **Further reply to Reviewer aBa7**
> > >
> > > Thank you for raising the score. Here are our responses to your minor concerns on the inference time comparison.
> > >
> > > We would like to point out that improving DEQs with some techniques like JFNK and DFSane is out of the scope of our work. To the best of our knowledge, there are currently no DEQs applying JFNK or DFSane methods. We cannot implement such comparisons easily. In this case, we believe our current experiments are quite standard and cannot be claimed "unfair".
> > >
> > > To avoid unnecessary misleading, in the final version, we will highlight that the DEQs implemented in our experiments are based on the Broyden solver.

---

### Official Review · Reviewer_rMvZ · 2023-07-06

**Soundness:** 3 good
**Presentation:** 2 fair
**Contribution:** 3 good
**Rating:** 5
**Confidence:** 3

**Summary:**

The work shows the connection between Deep Equilibrium Moldes (DEQs) and Neural Ordinary Differential Equations (NODEs) using homotopy continuation, a method to solve nonlinear equations. Based on the analysis, the authors suggest a new implicit model, HomoODE, which has high expressible power from DEQs and stability from NODEs. The method is demonstrated to obtain the best performance for image classification tasks between the models tested.

**Strengths:**

* The connection between DEQ and NODE is presented clearly and seems mathematically solid, which delivers a lot of insights to the community (in particular, in the field of implicit models).
* In addition to mathematical notions, the paper covers practical ideas, e.g., the learnable initial condition. The effect of the adjoint method is also investigated.

**Weaknesses:**

* The reason why the authors choose image classification is not very clear. With the current manuscript, the practical contribution of the paper looks quite limited because DEQ is designed for sequential data, and NODE is for ODEs. Thus, the reviewer cannot assess the effectiveness of HomoODE because image classification is not a main field of the baseline methods. The reviewer strongly recommends the authors run experiments on ODEs to show real improvement compared to DEQ and NODE.
* Related to the first point, the reason why the HomoODE works better in the image classification task is not clear. The reviewer did not find out what the authors would like to state with these experiments. I believe there are a lot of excellent models to compare in the field of image classification if the authors would like to claim the model's effectiveness in the image classification tasks. It could be understandable if the image classification task is to show the adaptability of the model, and there is another main experiment (e.g., on an ODE dataset).

**Questions:**

* How does one determine $v$? Is it a hyperparameter or computed using some equation?
* What do "explicit" and "implicit" mean in, e.g., "unlike DEQs, which explicitly solve equilibrium-point-finding problems via Newton's methods, our HomoODE solves these problems based on homotopy continuation implicitly"? It could be unclear and require some explanation because the meaning of "implicit" seems different from that of "implicit NN."
* What are "two initial conditions" in "there might be no solution for $\lambda(t)$ as there are two initial conditions for the system"?
* What do $h$, $w$, and $c$ stand for in line 265?
* Why is the inference time of HomoODEs with data augmentation longer than those without data augmentation in Table 1 (e.g., HomoODE vs. HomoODE$^*$, or HomoODE$^\dagger$ vs. HomoODE$^{*\dagger}$)? Is it due to increased iterations due to training with the augmented dataset?
* Could you please elaborate on the meaning of "fixed condition" in "Neural ODEs solve an equilibrium-point-finding problem with fixed condition $x$ and different initial points $z(t_0)$"? The statement seems to contradict because Equation (5) states that $z(t_0) = x$, so different $z(t_0)$ implies different $x$. In addition, Neural ODE can take $t$ as an input, which evolves in time.

**Limitations:**

* The method to optimize the shared initial information seems tricky and could be stated as a part of the limitation.

---

> ### Author Rebuttal · Authors · 2023-08-09
>
> We would like to thank the Reviewer rMvz for the detailed and constructive comments. In the following, we have provided an item-by-item response to the comments.
>
> > **Q1**: The reason why the authors choose image classification is not very clear. The practical contribution looks quite limited because DEQ is designed for sequential data, and NODE is for ODEs. The reviewer strongly recommends the authors run experiments on ODEs to show real improvement compared to DEQ and NODE.
>
> **A1**: Thanks for the comments. Although DEQs were first designed for sequential data and Neural ODEs were for ODEs, image datasets have been frequently used in the subsequential works [7, 43, 10, 12, R1, R2, R3] of DEQs and Neural ODEs. The main target of this paper is to establish the connection between DEQs and Neural ODEs via homotopy continuation. **Choosing suitable datasets for both DEQs and Neural ODEs is important**. Since ODE datasets are not suitable for DEQs, so we cannot compare on this dataset. Sequential datasets are not suitable as well due to the inconsistent backbones. For sequential datasets, DEQ models are based on transformer, while Neural ODEs are based on fully-connected layers. For image datasets, both DEQs and Neural ODEs use standard convolutional layers. Therefore, we consider image classification datasets as a reasonable and appropriate choice.
>
> [R1] Li, Mingjie, et al. "Optimization inspired multi-branch equilibrium models." ICLR 2021.
>
> [R2] Pokle, Ashwini, Zhengyang Geng, and J. Zico Kolter. "Deep equilibrium approaches to diffusion models." NeurIPS 2022.
>
> [R3] Zhuang, Juntang, et al. "Adaptive checkpoint adjoint method for gradient estimation in neural ode." ICML 2020.
>
> > **Q2 (1)**: Why the HomoODE works better in the image classification task is not clear.
>
> **A2(1)**: We believe the superiority of HomoODE comes from that HomoODE inherits the property of high accuracy from DEQs and avoid the instability issues of DEQs. Specifically, the structure of HomoODE enable it to solve the equilibrium point through the ODE solver, so it inherits the expressible representation capability from DEQs, and the property of stable convergence from Neural ODEs (line 224-227). The performance of DEQs is compromised by instability. Therefore, after handling the instability issue of DEQs, the performance of HomoODE improves accordingly.
>
> > **Q2(2)**: The reviewer believes there are a lot of excellent models to compare for image classification if the authors would like to claim the model's effectiveness in this task. It could be understandable if the image classification task is to show the adaptability of the model, and there is another main experiment (e.g., on an ODE dataset).
>
> **A2(2)**: We would like to clarify that we do not want to claim the model's effectiveness and adaptability, specifically in image classification tasks. Instead, we want to bridge DEQs and Neural ODEs via homotopy continuation. Therefore, we need to choose suitable datasets for both DEQs and Neural ODEs. As we illustrated in **A1**, image classification tasks play a fair role for both DEQs and Neural ODEs and the ODE datasets are unsuitable for DEQ models.
>
>
> >**Q3**: How does one determine $v$? Is it a hyperparameter or computed using some equation?
>
> **A3**: $v$ is neither a hyperparameter nor computed using some equation. Instead, it is an auxiliary variable for the theoretical analysis to bridge the DEQs and Neural ODEs via Fixed Point Homotopy (line 178-184).
>
> >**Q4**: What do "explicit" and "implicit" mean in "DEQs explicitly solve equilibrium-point-finding problems via Newton's methods, our HomoODE solves these problems based on homotopy continuation implicitly"? Require explanation since "implicit" seems different from "implicit NN".
>
> **A4**: The "implicitly" in this sentence is indeed different from that of "implicit NN" as DEQs and Neural ODEs. The "explicit" in this sentence means DEQs solve the underlying equilibrium-point-finding problem $z^* = f(z^*;x,\theta)$ explicitly via Newton’s methods. In HomoODE, we cannot directly obtain the form of its underlying equilibrium-point-finding problem $f(z;x,\theta)$, so we claim "our HomoODE solves these problems based on homotopy continuation implicitly”. We will add these analyses in the final version.
>
> >**Q5**: What are "two initial conditions" in "there might be no solution for $\lambda(t)$ as there are two initial conditions for the system"?
>
> **A5**: The two initial conditions are $\lambda(0) = 0$ and $\lambda(1) = 1$ (Equation 11), which come from the theory of homotopy continuation.
>
> >**Q6**: What do $h$, $w$, and $c$ stand for in line 265?
>
> **A6**: Here $h, w, c$ means the height, width and channel number of the feature map. We will add a more detailed description for them in our final version.
>
> >**Q7**: Why is the inference time of HomoODE with data augmentation longer than those without data augmentation in Table 1 (e.g., HomoODE vs. HomoODE$^*$, or HomoODE$^\dagger$ vs. HomoODE $^{\dagger*}$)? Is it due to increased iterations due to training with the augmented dataset?
>
> **A7**: Yes, the number of function evaluations (NFE) of HomoODE increased with the augmented dataset. HomoODE learns better representations through augmented datasets, resulting in increased NFE and higher test accuracy, which leads to longer inference time.
>
> >**Q8**: Elaborate the meaning of "fixed condition" in "Neural ODEs solve an equilibrium-point-finding problem with fixed condition $x$ and different initial points $z(t_0)$"? The statement seems to contradict ....
>
> **A8**: Thank you for pointing out this. There is a typo in this sentence, and it should be "Neural ODEs solve the fixed equilibrium-point-finding problem with different initial points $z(t_0)$". There is no condition in Neural ODEs, and here we want to state the underlying equilibrium-point-finding problem of Neural ODEs does not vary with the input. We will modify this statement in the final version.

---

> > ### Comment · Reviewer_rMvZ · 2023-08-16
> >
> > Thank you for the response. Overall, the authors addressed most of the concerns raised by the first review. Therefore, I raised the score (contribution 2 -> 3, rating 4 -> 5). However, the reviewer would like the authors to clarify the reason why "ODE datasets are not suitable for DEQs." In addition, if only the image classification task is suitable for the fair comparison of NODE and DEQ (and HomoODE), the reviewer is confused about the benefit of bridging NODE and DEQ. I initially considered that HomoODE might be useful if it incorporate the expressibility of DEQ into ODE for ODE tasks for example, but is it wrong?

---

> > > ### Author Response · Authors · 2023-08-18
> > > **Reply to Reviewer rMvZ**
> > >
> > > Thank you for raising the score. Here are our responses to your minor concerns.
> > > > The reviewer would like the authors to clarify the reason why "ODE datasets are not suitable for DEQs."
> > >
> > > We suppose the so-called "ODE datasets" mentioned by the reviewer is about the simulation tasks of continuous physical process. DEQs inherently are not designed to handle this type of task as DEQs do not involve the continuous time series in the models. Although DEQs based on Transformers are applied in sequential tasks [RR1], they are still discrete and are not suitable for the continuous physical process.
> > >
> > > >  In addition, if only the image classification task is suitable for the fair comparison of NODE and DEQ (and HomoODE), the reviewer is confused about the benefit of bridging NODE and DEQ. I initially considered that HomoODE might be useful if it incorporate the expressibility of DEQ into ODE for ODE tasks for example, but is it wrong?
> > >
> > > We believe our HomoODE is still useful for Neural ODEs. Generally, Neural ODEs are used to achieve two different purposes.
> > > * One is to treat Neural ODEs as a memory-efficient feature extractor. Research works for this purpose often test Neural ODEs on image classification tasks [RR2, RR3]. Notably, as a well-known ODE paper, Augmented Neural ODE [RR2] and Neural SDE [RR3] only considered the image classification tasks as well.
> > > * The other is to simulate a continuous physical process and then make predictions or decisions for these tasks. For example, Rubanova et. al [RR4] only considered these tasks without image classification ones.
> > >
> > > **Our HomoODE does improve the representation capability of Neural ODEs from the perspective of the first purpose by providing
> > > a not only memory-efficient but also representative feature extractor**.
> > >
> > > [RR1] Bai, S., Kolter, J. Z., & Koltun, V. (2019). Deep equilibrium models. Advances in Neural Information Processing Systems, 32.
> > >
> > > [RR2] Dupont, Emilien, Arnaud Doucet, and Yee Whye Teh. "Augmented neural ODEs." Advances in neural information processing systems 32 (2019).
> > >
> > > [RR3] Liu X, Xiao T, Si S, et al. How does noise help robustness? explanation and exploration under the neural SDE framework[C]//Proceedings of the IEEE/CVF Conference on Computer Vision and Pattern Recognition. 2020: 282-290.
> > >
> > > [RR4] Rubanova, Y., Chen, R. T., & Duvenaud, D. K. (2019). Latent ordinary differential equations for irregularly-sampled time series. Advances in neural information processing systems, 32.

---

> > > > ### Comment · Reviewer_rMvZ · 2023-08-18
> > > >
> > > > Thank you for the clarification!

---

### Author Rebuttal · Authors · 2023-08-09

We thank the reviewers for their careful reading and constructive and detailed suggestions to help us improve our manuscript.  We sincerely appreciate that the reviewers consider unanimously that our work "effectively" (ZDnx) establishes an "inherent and insightful" (iBob) and "mathematically solid" (rMvZ) connection in the absence of theoretical explorations between Neural ODEs and DEQs (aBa7). We are further glad that the reviewers agree our work provides "nice" and "practical" (rMvZ, iBob) ideas like the learnable initial point, and that the proposed HomoODE inherits the advantages of DEQs and Neural ODEs (iBob, ZDnx), which delivers a lot of insights and contributes significantly to the community of implicit model (iBob, rMvZ).

In the following, we will try to address the concerns/questions of the reviewers and provide a detailed item-by-item response to your comments.

---

### Decision · Program_Chairs · 2023-09-21

**Decision:**

Accept (poster)

**Comment:**

The reviewers unanimously find the paper interesting and result worth publication. The area chair agrees with the evaluation after reading the discussion and the manuscript.